# The Origins and Future of Sentinel: An Early-Warning System for Pandemic Preemption and Response

**DOI:** 10.3390/v13081605

**Published:** 2021-08-13

**Authors:** Yolanda Botti-Lodovico, Parvathy Nair, Dolo Nosamiefan, Matthew Stremlau, Stephen Schaffner, Sebastian V. Agignoae, John Oke Aiyepada, Fehintola V. Ajogbasile, George O. Akpede, Foday Alhasan, Kristian G. Andersen, Danny A. Asogun, Oladele Oluwafemi Ayodeji, Aida S. Badiane, Kayla Barnes, Matthew R. Bauer, Antoinette Bell-Kareem, Muoebonam Ekene Benard, Ebo Ohomoime Benevolence, Osiemi Blessing, Chloe K. Boehm, Matthew L. Boisen, Nell G. Bond, Luis M. Branco, Michael J. Butts, Amber Carter, Andres Colubri, Awa B. Deme, Katherine C. DeRuff, Younousse Diédhiou, Akhilomen Patience Edamhande, Siham Elhamoumi, Emily J. Engel, Philomena Eromon, Mosoka Fallah, Onikepe A. Folarin, Ben Fry, Robert Garry, Amy Gaye, Michael Gbakie, Sahr M. Gevao, Gabrielle Gionet, Adrianne Gladden-Young, Augustine Goba, Jules Francois Gomis, Anise N. Happi, Mary Houghton, Chikwe Ihekwuazu, Christopher Ojemiega Iruolagbe, Jonathan Jackson, Simbirie Jalloh, Jeremy Johnson, Lansana Kanneh, Adeyemi Kayode, Molly Kemball, Ojide Chiedozie Kingsley, Veronica Koroma, Dylan Kotliar, Samar Mehta, Hayden C. Metsky, Airende Michael, Marzieh Ezzaty Mirhashemi, Kayvon Modjarrad, Mambu Momoh, Cameron A. Myhrvold, Okonofua Grace Naregose, Tolla Ndiaye, Mouhamadou Ndiaye, Aliou Ndiaye, Erica Normandin, Ikponmwosa Odia, Judith Uche Oguzie, Sylvanus A. Okogbenin, Peter O. Okokhere, Johnson Okolie, Idowu B. Olawoye, Testimony J. Olumade, Paul E. Oluniyi, Omigie Omoregie, Daniel J. Park, Mariétou Faye Paye, Brittany Petros, Anthony A. Philippakis, Abechi Priscilla, Alan Ricks, Anne Rimoin, John Demby Sandi, John S. Schieffelin, Monica Schreiber, Mame Cheikh Seck, Sameed Siddiqui, Katherine Siddle, Allison R. Smither, Mouhamad Sy, Ngayo Sy, Christopher H. Tomkins-Tinch, Oyewale Tomori, Chinedu Ugwu, Jessica N. Uwanibe, Eghosasere Anthonia Uyigue, Dada Ireti Victoria, Anika Vinzé, Megan E. Vodzak, Nicole Welch, Haja Isatta Wurie, Daba Zoumarou, Donald S. Grant, Daouda Ndiaye, Bronwyn MacInnis, Pardis C. Sabeti, Christian Happi

**Affiliations:** 1Broad Institute of Massachusetts Institute of Technology (MIT) and Harvard, Cambridge, MA 02142, USA; ybottilo@broadinstitute.org (Y.B.-L.); sfs@broadinstitute.org (S.S.); sagignoa@broadinstitute.org (S.V.A.); kbarnes@broadinstitute.org (K.B.); cboehm@broadinstitute.org (C.K.B.); mjbutts@broadinstitute.org (M.J.B.); acarter@broadinstitute.org (A.C.); andres@broadinstitute.org (A.C.); kderuff@broadinstitute.org (K.C.D.); selhamou@broadinstitute.org (S.E.); ggionet@broadinstitute.org (G.G.); agladden@broadinstitute.org (A.G.-Y.); mhoughto@broadinstitute.org (M.H.); jjohnson@broadinstitute.org (J.J.); mkemball@broadinstitute.org (M.K.); dkotliar@broadinstitute.org (D.K.); hmetsky@broadinstitute.org (H.C.M.); marzieh@broadinstitute.org (M.E.M.); cmyhrvol@princeton.edu (C.A.M.); enormand@broadinstitute.org (E.N.); dpark@broadinstitute.org (D.J.P.); mpaye@broadinstitute.org (M.F.P.); bpetros@broadinstitute.org (B.P.); aphilipp@broadinstitute.org (A.A.P.); ssiddiqu@broadinstitute.org (S.S.); kjsiddle@broadinstitute.org (K.S.); tomkinsc@broadinstitute.org (C.H.T.-T.); avinze@broadinstitute.org (A.V.); mvodzak@broadinstitute.org (M.E.V.); nwelch@broadinstitute.org (N.W.); bronwyn@broadinstitute.org (B.M.); 2Howard Hughes Medical Institute, Chevy Chase, MD 20815, USA; pnair@broadinstitute.org; 3African Center of Excellence for Genomics of Infectious Diseases (ACEGID), Redeemer’s University, Ede, Osun State, Nigeria; dolonosa@broadinstitute.org (D.N.); ajogbasilef@run.edu.ng (F.V.A.); 2mimijoe@gmail.com (P.E.); folarino@run.edu.ng (O.A.F.); anisehappi@yahoo.com (A.N.H.); kayodet@run.edu.ng (A.K.); oguziej@run.edu.ng (J.U.O.); okoliec@run.edu.ng (J.O.); olawoyei0303@run.edu.ng (I.B.O.); olumadet@run.edu.ng (T.J.O.); oluniyip@run.edu.ng (P.E.O.); abechip@run.edu.ng (A.P.); ugwuc@run.edu.ng (C.U.); uwanibej@run.edu.ng (J.N.U.); dadaireti@gmail.com (D.I.V.); 4Equator Labs Incorporated, Washington, DC 20011, USA; stremlau@gmail.com; 5Department of Organismic and Evolutionary Biology, Harvard University, Cambridge, MA 02138, USA; mschreib@broadinstitute.org; 6Department of Immunology and Infectious Diseases, Harvard T.H. Chan School of Public Health, Harvard University, Boston, MA 02115, USA; 7Harvard Kennedy School, Cambridge, MA 02138, USA; 8Institute of Lassa Fever, Research and Control, Irrua Specialist Teaching Hospital, Irrua, Edo State, Nigeria; aiyepadaoke82@yahoo.com (J.O.A.); georgeakpede@yahoo.co.uk (G.O.A.); nwata2007@gmail.com (M.E.B.); ebobene77@yahoo.com (E.O.B.); blessingosiemi@rocketmail.com (O.B.); patienceedamhande@yahoo.com (A.P.E.); chikwe.ihekweazu@gmail.com (C.I.); airemikelis@gmail.com (A.M.); grace4ambi84@gmail.com (O.G.N.); iodia905@gmail.com (I.O.); okogbenins@yahoo.com (S.A.O.); pitaokokhere@yahoo.com (P.O.O.); oomigie@ymail.com (O.O.); oyewaletomori@gmail.com (O.T.); diamondeghe@gmail.com (E.A.U.); 9Department of Biological Sciences, Redeemer’s University, Ede, Osun State, Nigeria; daouda.ndiaye@ucad.edu.sn; 10Department of Medicine, Faculty of Clinical Sciences, College of Medicine, Ambrose Alli University, Ekpoma, Edo State, Nigeria; 11Viral Hemorrhagic Fever Program, Kenema Government Hospital, Ministry of Health and Sanitation, Kenema, Sierra Leone; fodayalhasan37@gmail.com (F.A.); gbakiemichael@gmail.com (M.G.); lansanakanneh@gmail.com (L.K.); vjkoroma@gmail.com (V.K.); donkumfel@yahoo.co.uk (D.S.G.); 12Department of Immunology and Microbiology, The Scripps Research Institute, La Jolla, CA 92037, USA; andersen@scripps.edu; 13Scripps Research Translational Institute, La Jolla, CA 92037, USA; 14Ambrose Alli University, Ekpoma, Edo State, Nigeria; asogun2001@yahoo.com; 15Federal Medical Centre, Owo, Ondo State, Nigeria; femiayodeji@yahoo.com; 16Université Cheikh Anta Diop, BP 5005, Dakar, Senegal; asbadiane@gmail.com (A.S.B.); deme.awa@gmail.com (A.B.D.); ydjedju@yahoo.fr (Y.D.); myanaa08@gmail.com (A.G.); jules.gomis@gmail.com (J.F.G.); ndiayetola@gmail.com (T.N.); mouhamadou.ndiaye@ucad.edu.sn (M.N.); aliou.ndiaye10@yahoo.fr (A.N.); mcseck203@yahoo.fr (M.C.S.); symouhamad92@gmail.com (M.S.); ngayosy50@hotmail.com (N.S.); rwallisz@yahoo.fr (D.Z.); 17Division of Medical Sciences, Harvard Medical School, Boston, MA 02115, USA; mbauer@broadinstitute.org; 18Tulane University Medical Center, New Orleans, LA 70112, USA; abell13@tulane.edu (A.B.-K.); nbond@tulane.edu (N.G.B.); eengel@tulane.edu (E.J.E.); rfgarry@tulane.edu (R.G.); 19Department of Molecular Biology, Princeton University, Princeton, NJ 08540, USA; 20Zalgen Labs, Germantown, MD 20876, USA; mboisen24@gmail.com (M.L.B.); lbranco@zalgenlabs.com (L.M.B.); 21University of Massachusetts Medical School, Worcester, MA 01605, USA; 22Refuge Place International, Bassa Town, Lower Johnsonville, Liberia; mfallah1969@gmail.com; 23Fathom Information Design, Boston, MA 02114, USA; ben@fathom.info; 24University of Sierra Leone, Freetown, Sierra Leone; gevaosm@yahoo.com; 25College of Medicine and Allied Health Sciences, University of Sierra Leone, Freetown, Sierra Leone; augstgoba@yahoo.com (A.G.); simbiriej5@gmail.com (S.J.); mambumomoh@gmail.com (M.M.); johnatsandi@gmail.com (J.D.S.); imwurie@icloud.com (H.I.W.); 26Department of Internal Medicine, Irrua Specialist Teaching Hospital, Irrua, Edo State, Nigeria; chriojem@gmail.com; 27Dimagi, Inc., Cambridge, MA 02139, USA; jjackson@dimagi.com; 28Department of Medical Microbiology, Alex-Ekwueme Federal University Teaching Hospital Abakaliki, Abakaliki, Ebonyi, Nigeria; edomann2001@yahoo.com; 29Department of Critical Care Medicine, University of Maryland Medical Center, Baltimore, MA 21201, USA; sbmehta@gmail.com; 30Walter Reed Army Institute of Research, Silver Spring, MD 20910, USA; kmodjarrad@eidresearch.org; 31Eastern Polytechnic College, Kenema, Sierra Leone; 32Department of Systems Biology, Harvard Medical School, Boston, MA 02115, USA; 33Department of Medicine, Irrua Specialist Teaching Hospital, Irrua, Edo State, Nigeria; 34West African Examinations Council, Yaba, Lagos State, Nigeria; 35Harvard-MIT Health Sciences and Technology, Cambridge, MA 02139, USA; 36Harvard/MIT MD-PhD Program, Boston, MA 02115, USA; 37MASS Design Group, Boston, MA 02116, USA; alan@mass-group.org; 38Yale School of Architecture, New Haven, CT 06511, USA; 39Department of Epidemiology, Jonathan and Karing Fielding School of Public Health, University of California, Los Angeles, CA 90095, USA; arimoin@g.ucla.edu; 40Section of Infectious Disease, Department of Pediatrics, Tulane University School of Medicine, New Orleans, LA 70112, USA; jschieff@tulane.edu; 41Computational and Systems Biology Program, Massachusetts Institute of Technology, Cambridge, MA 02139, USA; 42Department of Microbiology and Immunology, Tulane University School of Medicine, New Orleans, LA 70112, USA; asmither@tulane.edu; 43Ikorodu General Hospital, Ikorodu, Lagos State, Nigeria; 44Department of Medicine, Division of Infectious Diseases, Massachusetts General Hospital, Boston, MA 02114, USA; 45Massachusetts Consortium on Pathogen Readiness, Boston, MA 02115, USA

**Keywords:** pandemic preemption, pandemic response, diagnostic tools, bioinformatics, genomic surveillance, infectious disease, Lassa virus, Lassa fever, Ebola, LARGE

## Abstract

While investigating a signal of adaptive evolution in humans at the gene LARGE, we encountered an intriguing finding by Dr. Stefan Kunz that the gene plays a critical role in Lassa virus binding and entry. This led us to pursue field work to test our hypothesis that natural selection acting on LARGE—detected in the Yoruba population of Nigeria—conferred resistance to Lassa Fever in some West African populations. As we delved further, we conjectured that the “emerging” nature of recently discovered diseases like Lassa fever is related to a newfound capacity for detection, rather than a novel viral presence, and that humans have in fact been exposed to the viruses that cause such diseases for much longer than previously suspected. Dr. Stefan Kunz’s critical efforts not only laid the groundwork for this discovery, but also inspired and catalyzed a series of events that birthed Sentinel, an ambitious and large-scale pandemic prevention effort in West Africa. Sentinel aims to detect and characterize deadly pathogens before they spread across the globe, through implementation of its three fundamental pillars: Detect, Connect, and Empower. More specifically, Sentinel is designed to detect known and novel infections rapidly, connect and share information in real time to identify emerging threats, and empower the public health community to improve pandemic preparedness and response anywhere in the world. We are proud to dedicate this work to Stefan Kunz, and eagerly invite new collaborators, experts, and others to join us in our efforts.

## 1. Background

In 2007, our team set out to investigate an intriguing hypothesis, catalyzed by the research of Dr. Stefan Kunz, about the potentially ancient origins of Lassa virus. Lassa virus is the cause of a viral hemorrhagic fever known as Lassa fever and a category A bioterror threat. Despite its deadly nature and designation as an “emerging” infectious disease, we had reason to believe, based on our investigations of the human genome, that Lassa virus had circulated for millennia in Nigeria, conferring a selective pressure on human populations. Our work pursuing this hypothesis has led to a multidisciplinary, multinational viral genomics research program, and eventually our creation of *Sentinel*: an ambitious and innovative pandemic prevention effort now being piloted in West and Central Africa. Here we describe the history and principle behind Sentinel.

It began as a signal of human adaptation, connected to a gene called LARGE. As part of a genome survey of human variation in Europe, Asia, and Africa with the International Haplotype Map Consortium, Dr. Pardis Sabeti’s team had developed methods for investigating signals of recent (within the last ~10 k years) natural selection in the human genome [1]. The team identified the locus surrounding LARGE as the strongest signal of recent selection in the survey; the signal was only visible in the African portion of the data, which came from the Yoruba people in Nigeria. From the literature, we learned that LARGE is a glycosylase that post-translationally modifies α-dystroglycan. This led us to a paper by Kunz, Michael Oldstone, and colleagues showing that LARGE’s modification of α-dystroglycan, the cellular receptor for Lassa virus, was critical for efficient virus binding and entry to mammalian cells [2,3]. Knocking out the gene encoding LARGE greatly tempered Lassa virus’s ability to infect cells. Taken together, these observations raised the possibility that polymorphisms in LARGE were under positive selection because they disrupted the protein’s function, thereby preventing Lassa virus entry into host cells and conferring resistance to Lassa fever. This realization further led us to hypothesize that Lassa virus may in fact have ancient origins, but circulated unnoticed in part due to genetic resistance to the virus in populations where Lassa fever is endemic.

To test this hypothesis, we had to collect fundamental information about the virus such as its age, geographic location, as well as the range and length of time that it had been circulating in affected zones. In particular, since the common variant in LARGE appeared to arise within the last 10,000 years, dating the age of the virus was essential to identifying the virus as a potential driver of natural selection for the gene.

We obtained the first pieces of evidence to support our hypothesis when we learned that the first geographic location in which Lassa fever was initially discovered was Nigeria in 1969 [4], where the signal of selection was also detected. We also learned that Lassa fever is endemic to West Africa [5]. In subsequent investigations of other global populations, we found that the signal of natural selection at LARGE was only present among populations in which Lassa fever was known to be present [6]. Today, Lassa fever is estimated to afflict 100,000–300,000 people each year, hospitalize 100,000, and kill 5000, although diagnostic testing remains limited and estimates vary widely [7]. Intriguingly, however, natural resistance to Lassa fever in parts of West Africa is common, with 50–90% of individuals showing mild or no symptoms when infected, while others suffer a complicated course that can lead to fever, encephalitis, deafness, and death [5]. It was unknown, however, why Lassa fever infection in some individuals was asymptomatic, while others developed severe disease.

To examine differential susceptibility to Lassa fever in individuals, the age and diversity of the virus, and whether Lassa virus was driving natural selection for thousands of years, we would need to study the genomes of both the virus and patients affected by it. We believed that with new tools to assay genetic variation across the human genome, we could examine and characterize any genetic evidence of resistance to Lassa fever, a study which we then set out to perform.

## 2. Studying Lassa Fever in Nigeria

Investigating a poorly understood, category A virus in rural Nigeria would require us to overcome many challenges. Study of Lassa virus and other high containment (biosafety level (BSL) 4) pathogens is complicated due to their lethality, lack of effective treatment, potential for aerosol transmission, and bioterror use, requiring rigorous precautions by highly trained teams. Moreover, at the time of our research, these diseases often appeared as sporadic outbreaks in remote settings, where infrastructure and transportation were lacking. However, with increased globalization and an ever-expanding human population, the need for large-scale research initiatives on BSL-4 pathogens remains acute [8,9]. Further, as only one BSL-4 lab exists in the entire region of West Africa [10] even today, transnational partnerships are critical to allow ongoing investigation of BSL-4 pathogen samples.

In order to study Lassa virus, Dr. Sabeti teamed up with Dr. Christian Happi, a colleague from previous studies on malaria, who had just begun doing viral research in Edo State, Nigeria, in the heart of the Lassa fever endemic zone in Nigeria. The nearby Irrua Specialist Teaching Hospital (ISTH) in Irrua, Nigeria was also the site of an upcoming government-funded center of excellence for Lassa fever, which at the time did not have diagnostics or treatment available onsite for the virus. Together, Happi and Sabeti’s team established a partnership with ISTH in 2008, to develop diagnostics capacity onsite and facilitate acquisition of Ribavirin, the WHO essential therapeutic for Lassa fever.

ISTH treats ~16,000 patients each year, 80% of whom suffer from febrile illness [11,12]. The region suffers annual outbreaks of Lassa fever, but due to the lack of diagnostic tools or local capacity for surveillance, as well as the inaccessibility of clinics in remote regions, most local Lassa fever cases had gone undetected or were misdiagnosed as similarly presenting acute febrile illnesses. Drs. Happi and Sabeti worked alongside the Bernard Nocht Institute to establish on-site PCR diagnostics able to detect Lassa virus in patient blood, and to help source Ribavirin for the hospital. Rapid and accurate diagnostics led to more informed treatment plans for patients with Lassa fever, which facilitated an overall improvement in community engagement and health outcomes. The surrounding communities began hearing many stories of successful Lassa fever detection and recovery, and the site quickly became a referral center for patients with undiagnosed febrile illness within hundreds of kilometres.

Early in our work, we also gleaned an important insight after a man brought his young son to the hospital, where doctors diagnosed him with Lassa fever and successfully treated him. During a follow up investigation into the prevalence of Lassa virus in the patient’s village, Dr. Happi learned that the child’s mother, cousin, and neighbor had all succumbed to what appeared to be the same mysterious illness. In total, more than 20 members of that community who had died exhibited classical symptoms of clinical Lassa fever, without inciting any alarm or broader action by the public health community. One explanation for this may have been the fact that none of the deceased sought help from the nearby hospital, which previously lacked both diagnostic tests for Lassa fever, and sufficient capacity to treat the disease. Overall, the experience led us to contemplate the countless individuals who succumb to Lassa fever and other underdiagnosed diseases [13]. 

Based on findings from genetic and epidemiologic studies, we wrote a perspective entitled ‘Emerging Disease or Diagnosis?’ suggesting that we wrongly label some viruses as novel or rare, not because they truly are, but because we lack the necessary tools in the relevant places to detect them [9]. Our analysis of Lassa virus genomes suggested that the virus had been circulating for at least half a millenia in West Africa [14]. Contrary to popular belief, viral hemorrhagic fevers often exhibit nonspecific symptoms, like fever, nausea, headache, and malaise, which can complicate diagnosis. Without access to accurate diagnostics, healthcare workers are stymied in determining the etiology of fever, and many wrongly presume that malaria, typhoid, or shigella is responsible [9]. In the worst case scenario, lives are lost as a result of misdiagnosis, as in a 1989 Lassa fever outbreak in two Nigerian hospitals, which took the lives of 22 people [15]. This work led us to the realization that in many parts of the world, we are largely blind both to the prevalence of known infectious diseases and to the appearance of new threats. As we develop and equip local healthcare workers with proper diagnostic tools, however, we can detect areas of higher prevalence and subsequently develop more effective treatments, vaccines, surveillance tools, and research capacity where they are most needed, instead of awaiting the next outbreak.

## 3. ACEGID: Laying the Groundwork for Local Surveillance and Outbreak Response

Like Kunz, we began our work on Lassa virus with the goal of conducting molecular biology research on the mechanisms behind viral transmission and human resistance to Lassa fever disease. But to enable this research, we quickly recognized that responding to the basic diagnostic needs on the front line would be critical. More broadly, we also recognized that a sustained partnership between communities and their local healthcare clinics establishes trust, while the improved capacity of healthcare workers to detect and treat common diseases encourages more individuals to seek tests when needed, thereby increasing the number of samples available for important research [9]. 

As we continued to work with our partners at ISTH in Nigeria—and later extending to the Kenema Government Hospital (KGH) in Sierra Leone—to set up affordable diagnostics for routine use, we were able to establish this positive feedback loop, in which the higher number of diagnostic samples we collected at the point of care enabled us to conduct genomic sequencing efforts to identify and characterize other circulating viruses in the region. As new viral threats emerged, we were then able to quickly establish accurate diagnostics onsite to better serve communities. Since then, ISTH and KGH have become reference centers for management of viral hemorrhagic fevers in their respective regions. They have successfully detected more cases and saved more lives, thereby raising national awareness around these diseases. For example, in 2012 the Nigerian Federal Ministry of Health declared an increase in the number of suspected Lassa fever cases, an example of a possible “emergence of diagnosis”; with nearly 1000 cases reported by 41 local government agencies in 23 States [9]. These reports enabled rapid response measures by the government and public health community.

By standardizing and implementing this virtuous cycle as a broader system, we can ultimately build a much needed global surveillance capacity, strengthened by ongoing collaboration and trust between communities, local healthcare workers, hospitals like ISTH and KGH, and public health decision makers. These efforts have the potential to immediately benefit affected communities where certain diseases are endemic, while also helping to identify, monitor, and characterize emerging pathogens before they cause a regional outbreak or global pandemic.

As a first step in realizing this kind of collaborative surveillance model in West Africa, we merged these ideas into one blueprint, which laid the groundwork for our launch of the African Center of Excellence for Genomics of Infectious Disease (ACEGID). Based at Redeemer’s University in Ede, Nigeria, ACEGID has built a network of partners in West Africa, including ISTH, KGH, and the Université Cheikh Anta Diop de Dakar in Senegal, supported by partners at Tulane University, and funded by the NIH Human Heredity and Health Africa Initiative and the World Bank. Its mission is organized into five specific mandates, including: (1) building research capacity among African scientists in genomics and molecular biology; (2) empowering researchers to apply genomic tools and data toward containing and defeating infectious diseases; (3) generating a curriculum to promote and support high-quality genomics research for the purpose of improving global health; (4) fostering a vibrant, collaborative, international research community, dedicated to high-quality, relevant, ethical, and responsive genomic research; and (5) engaging individuals and communities in outbreak prevention, education, and local public health practice, while also supporting clinical care and building a surveillance network for global health threats. 

Just as we launched ACEGID in March 2014, however, Ebola began spreading through West Africa [8,16]. The outbreak went undiagnosed for months, reaching multiple countries, and mutating in ways that likely increased its infectivity [17,18,19]. By May, our KGH team, under the leadership of Dr. Humarr Khan, found itself on the frontlines of the local response efforts in Sierra Leone. We witnessed firsthand how a lack of proper diagnostics, particularly in remote areas, prevented healthcare workers from accurately determining the source of an infection, allowing it to spread far beyond local public health capacity. The outbreak eventually took 11,000 lives [16], including Dr. Khan and other members of our team who contracted Ebola while caring for patients. This served as a painful reminder of the deep need to detect and contain the broad range of deadly viruses that threaten the region every year. 

ACEGID went on to make critical contributions in the response to the Ebola outbreak, the Zero Malaria campaign, regional efforts to reduce the burden of Lassa fever and other infectious diseases, and more [20]. When the first case of Ebola arrived at the Lagos International Airport in 2014, our team at Redeemer’s diagnosed it, and helped stop a potentially devastating outbreak in its tracks [21]. Together, we have equipped over one thousand West African scientists with the education, training, and resources needed to monitor, study, and treat dangerous pathogens in their own region. 

## 4. A New Approach to Pandemic Preparedness

The success of ACEGID laid the foundation for Sentinel, a pandemic preemption system designed to detect and characterize deadly pathogens before they spread across the globe. This initiative, supported by the TED Audacious Prize, was funded one month before the declaration of the COVID-19 outbreak as a pandemic. 

Sentinel is organized around three core pillars, Detect, Connect, and Empower (Figure 1), that are as fundamental to preempting a pandemic as they are to responding to one that has already begun. More broadly, these pillars are interoperable and essential to supporting a public health system that can prevent a wide range of infectious diseases. The needs the pillars address remain as pressing today as they did when we conceived Sentinel, despite a year of vigorous public health effort to combat COVID-19. We describe the needs, as well as Sentinel’s response to them, in depth below.

## 5. Sentinel Pillar #1: Detect

Sentinel envisions a three-tiered approach to improving diagnosis and surveillance of viral infectious diseases. The first tier aims to detect the most likely pathogens at the point of care, using rapid, affordable diagnostic tests for the highest priority pathogens in a region. The second tier is designed to diagnose illnesses not covered by the point-of-care tests, and requires developing the capacity in local hospitals to identify a wide range of viruses in-house. The final tier relies on regional genomic sequencing centers to identify novel viruses and monitor known viruses; the resulting sequences can be useful for tracking changes to viruses, for monitoring spread of viral variants, and for developing new diagnostics and vaccines [22]. 

This approach places high demands on diagnostic technology. In the case of viral sequencing, the technology is already well-developed; the need is for creating sequencing capacity in places where it is currently unavailable, which is no easy task in itself. The technology can still benefit from enhancement, however, to increase scale, reduce cost, and improve sensitivity. The other tiers require new diagnostic technologies. The point-of-care tests, in particular, require the development of tests for many more viruses, tests that are rapid, affordable, and easy to use in settings with limited resources. Our team has already invested in a number of detection technologies to achieve these goals, focusing on CRISPR-based diagnostics that are sensitive, field-deployable, and rapidly programmable for a range of different pathogens. These CRISPR-based technologies can also be multiplexed to test for numerous viruses simultaneously in a single sample [23,24]. We are well aware, however, that the field of diagnostics is currently experiencing rapid changes: recent breakthroughs include novel point-of-care antigen capture technologies, nanotechnologies, and advances in isothermal amplification methods. Given the current pace of innovation, our goal is not to settle on a single technology in advance, but to have Sentinel serve as an innovation hub that can allow multiple technologies, including our own, to be piloted and optimized, retaining the flexibility to deploy the best technologies as they emerge. 

## 6. Sentinel Pillar #2: Connect

In addition to improving our diagnostic arsenal, Sentinel aims to address another key challenge in pandemic prevention: the ability to connect various sources of diagnostic and surveillance data, and to share that information in real time with those who need it to guide critical public health strategy. Through Sentinel we are building tools to address these needs at multiple levels, from symptom surveillance of individuals, to diagnostic and clinical data integration by healthcare workers, to genomic surveillance data. Ultimately, we envision a future in which all of these data streams are themselves connected in real time, providing a holistic picture of known and emerging threats. 

For years, our team has been developing tools and collaborating with others to make this vision a reality. Many of our solutions to data connectivity are cloud-based, mobile, and able to function in low connectivity environments. For example, the CommCare mobile app from Dimagi provides a means for frontline workers or individuals to share symptoms and diagnostic data and convey critical information between geographically separated clinics and labs in a way that is robust to intermittent and unreliable internet connectivity. Cloud genomic analysis platforms like Terra and DNAnexus bring strong compute resources and community-curated analysis pipelines to labs anywhere in the world, via a web-accessible interface. Powerful visualization tools of patient, epidemiologic, and genomic data sets, developed by Fathom, bring real time insights from field-generated data to those coordinating responses at the national level. 

Our strategy operates on a number of principles, including incentivizing the sharing of quality data, creating easy-to-use, participatory, and flexible systems for collecting and sharing data, and building community trust in the tools that inform outbreak response. By adapting these principles to the specific needs of each level, we aim to ensure a continuous ability to access and analyze each input of data. Given the resource limitations of the regions where Sentinel will operate, many of these tools are designed for use by scientists, laboratory staff, and public health workers with limited computational experience or on-site hardware resources. 

Ultimately, Sentinel’s aim is to integrate each of these tools to enable data movement across an interoperable and secure data ecosystem that empowers all of the stakeholders in the surveillance system. While our team is presently working to further develop and integrate our own existing suite of information tools, future efforts for integration of the broader global catalog of information tools will require new funding initiatives, as well as combined private and public sector efforts to expand Internet connectivity and mobile data service to remote regions across the globe.

## 7. Sentinel Pillar #3: Empower

Pandemic preemption requires diagnostic and informatics technologies, but the technology is futile unless it is used and the results integrated into clinical care and public health decision-making. A successful system must include every stakeholder, from government and public health officials, to frontline healthcare workers and scientists, to individual patients and communities. Distributed local capacity and individual agency are critical to stopping a pandemic. But limited funding, community awareness, and support for capacity building often prevent the broad and equitable deployment of existing tools and the timely development of new ones. 

Recognizing the importance of sustainability through local ownership, we regard training as paramount for building capacity in genomic sequencing, diagnostics, bioinformatics, and advanced genomic surveillance among local healthcare workers and laboratory scientists. Sentinel employs a train-the-trainers approach, organized through ACEGID, to generate sustainable and long-term capacity for use of Sentinel tools and implementation of each pillar. Leveraging our 10+ years of educational experience in West Africa, we aim to train 800 additional local healthcare workers and laboratory scientists to use our diagnostic and surveillance tools in pursuit of the Detect and Connect goals.

Sentinel also acknowledges the need for deep and longstanding partnerships to form integrated regional and global networks of experts and communities fully versed in pandemic preparedness and preemption. We know firsthand that this work is complex and challenging, and that our approach is only one of a myriad of efforts to support improved infectious disease diagnostics and surveillance. Having piloted this system in Nigeria and developed extensive partnerships through this work, we plan to grow our existing network and presence in Africa through cooperation with stakeholders like the Nigeria CDC, Africa CDC, and regional WHO offices in order to fulfill our role within this multi-stakeholder effort. As we anticipate the additional challenges that might arise over time, we will continually work to integrate into existing healthcare systems and empower local stakeholders, to understand and overcome reasons for resistance to our technologies, and to eventually build community trust, and encourage broad buy-in over time. Our proposed national innovation hubs in Nigeria, Senegal, and Sierra Leone serve as national genomics reference labs. The labs are closely connected to their respective national public health agencies and broader communities, which will ultimately drive rapid and strong local responses to outbreaks. 

## 8. Remembering Dr. Kunz in Our Work Today

Over one year into the COVID-19 pandemic, Sentinel’s focus on pandemic *preemption* continues to remain critically relevant. Dr. Stefan Kunz, whose groundbreaking work ignited our interest in Lassa fever, and led us onto a path towards new possibilities in genomic surveillance and our collective understanding of infectious disease, opened the potential for collaborative pandemic preemption efforts in Nigeria and beyond. There is still however much to do, and our path has not been a straightforward one. Our starting hypothesis, that variants in or around LARGE contribute to resistance to Lassa fever, remains an open question. We are still working to answer it, and to uncover other potential resistance variants. But the work has been complicated by multiple outbreaks, political instability, and violence at our hospitals, as well as the ongoing logistical challenges. Even as we have continued to pursue these efforts, we have also been responding to the more urgent priorities around infectious disease response on the ground. Nevertheless, as Sentinel matures, we are continuing on with this important research.

In the spirit of Dr. Kunz and his commitment to driving cutting-edge science and impactful partnerships for the improvement of global health, we remember the countless individuals who have worked alongside us, many of whom are still with us today, and some of whom we have tragically lost over this past decade. To honor their extraordinary contributions to public health and infectious disease research, we are proud to dedicate this work to Stefan Kunz, as well as our late colleagues who served on the frontlines of the Ebola response in Sierra Leone, including Alex Moigboi, Mohammed Fullah, Mbalu Fonnie, Vandi Sinnah, Alice Kovoma, S. Humarr Khan, Jacob Adu George Buanie, and Mohammed “Pa” Sow. Their tireless sacrifice, generosity of spirit, and unwavering dedication to their communities laid the groundwork for ongoing efforts to ensure that every individual has the right to health and quality care. As we continue to fight against infectious disease in West Africa and beyond, we remain grateful to them and the many partners who have made Sentinel possible, and we eagerly invite new collaborators, experts, and others to join us.

## Figures and Tables

**Figure 1 viruses-13-01605-f001:**
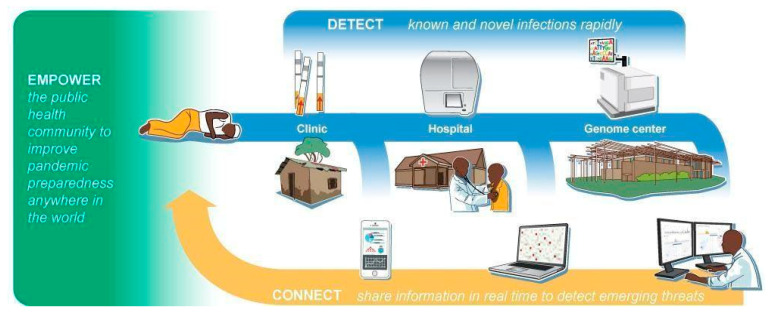
Sentinel will detect pathogens in any setting, from remote rural clinics to hospitals, and unknown pathogens will be identified and characterized at regional genome centers. Through a cloud-based system, Sentinel will share this information and connect healthcare workers, researchers, and disease control officials to track and predict threats. Finally, Sentinel will empower all stakeholders through a codified pandemic preemption system that can be scaled around the world.

## Data Availability

Data sharing is not applicable to this article.

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
