# Peer review of "The Origins and Future of Sentinel: An Early-Warning System for Pandemic Preemption and Response"

_viruses, 2021, doi:10.3390/v13081605_

Round 1

Reviewer 1 Report

This manuscript describes the inspiration based on Dr. Stefan Kunz’s work, and subsequent key events that contributed to the large-scale collaborative effort to enable the field study of Lassa virus in West Africa. Significant accomplishments including the establishment of partnerships with Irrua Specialist Teaching Hospital and Kenema Government Hospital, the launch of the African Center of Excellence for Genomic of Infectious Disease, and Sentinel project are discussed. This work is an excellent contribution to this special issue “In Memory of Stefan Kunz”.

Minor comments

Due to the collaborative nature of this project, the authorship represents a diverse group of institutions and viewpoints. When the first-person point of view is used in the manuscript, it is a bit confusing which groups are indicated. The authors should consider revising some of these statements for clarity.

There are several events described without a date. Including these would allow the reader to understand the timeline better.

Author Response

Attached below is the updated manuscript with the finalized text included as well as responses to the reviewers and a cover letter.

Reviewer 2 Report

The manuscript of Botti-Lodovico et al. describes the origins of Sentinel, a system aimed to prevent and minimize potential outbreaks of infectious diseases, such as LASV or EBOV. The idea was originally born from the identification of a potential selective trait in the LARGE locus, possibly associated to LASV infection resistance. This finding led the authors to hypothesize that emerging diseases could be misdiagnosed rather than emerging, which encouraged to the development of the Sentinel system. 

Although I found the manuscript interesting, I think there are some things that should be considered for its possible publication in Viruses:

-In their manuscript, the authors hypothesize that the signal detected surrounding LARGE locus could be associated to resistance to LASV infection. Nevertheless, the authors do not provide any direct experimental proof of such hypothesis nor compare the potential selection of the signal in human populations not exposed to LASV infection, which could serve as a reference for addressing this hypothesis.

-The authors described the origins of Sentinel, originally motivated by the findings of Dr. Kunz. Nevertheless, Sentinel aims for prevention and deeper study of LASV treatment, but there is, in my opinion, a missing narrative link between the Sentinel aims and the possible implications of Dr. Kunz findings into LASV field study. 

-The authors claims a supposed evolution towards higher infectivity of EBOV during its outbreak in 2014. To support this statement, the authors should cite relevant works.

-The authors hypothesize that LASV and other "emergent" diseases are more "unattended" rather "emergent". Again, besides the idea, the authors do not provide further supportive information. In this regard, although it is known that the majority of the LASV cases might be misdiagnosed or undetected, to support this hypothesis the authors should provide proofs that at least are suggestive of LASV infection or seroconversion in regions where LASV has not been previously detected.

-Although not absolutely required, an organigram of the Institutions/Organizations involved in the program would ease the reading and understanding of the manuscript. 

-The legend of Figure 1 is missing. 

-In the Tenet #1 (Detection) the authors highlight the importance of genomic sequencing. Although LASV genome sequencing is important tool and provides helpful information, is expensive and time consuming. Therefore, its use for early detection of LASV cases is rather discouraged. As authors also stated, appropriate quick and less demanding diagnostics seem to fulfill the early detection aim. If the authors maintain this statement, they should discuss how to use such technic in very short periods, required for the early detection of LASV.

-In the Tenet #2 (Empower) the authors describe a challenging scenario, which are the ways to connect and share information between the stakeholders of the system. Nevertheless, they fail to provide a solution for this issue. Similarly, in the Tenet #3, the authors do not describe how the coordination between the different elements in the system will occur. For a complete perspective of the Sentinel’s tenets, the authors should provide a plan to overcome the described challenges.

-In sum, although the manuscript highlight a very relevant program, its challenges, and achievements, I think a more detailed description of the plans to overcome the present and future challenges is required for its publication.

Author Response

(The authors gave the same response as above.)

Reviewer 3 Report

This paper by Botti-Lodivico and colleagues presents a concise but comprehensive and very clear description of the system Sentinel, wholistic system design to achieve early detection and control of human pathogens with the potential of triggering  serious outbreaks and, in the worst case scenario, a pandemic. 

The paper focuses on describing the steps required to successfully implement Sentinel in the context of Lassa virus, a mammarenavirus endemic to Western Africa that is responsible for a large number of Lassa fever (LF) cases yearly, a disease associated with high morbidity and significant mortality in hospitalized sever cases of LF.

The paper is well written and highly informative for those who are not experts in the area of epidemiology of infectious diseases. In addition, the authors have made a nice and sensitive link of their paper to the contributions made to the field by Stefan Kunz, who is being honored in this special issue of Viruses.

I have only some minor comments for the authors consideration. Although the role of alpha dystroglycan (aDG) as a host cell factor required for efficient LASV cell entry has been solidly established. There is a substantial body of evidence indicating the aDG is not strictly required for LASV cell entry, but rather for the efficiency of the virus cell entry process. This should be captured in the background section of the paper. 

It would be also important to make a clear distinction between viruses characterized by outbreaks, like Ebola virus, versus those that are endemic and constantly circulating such as LASV in Western Africa.

Author Response

(The authors gave the same response as above.)
